# The Impact of Internal Insulation on Heat Transport through the Wall: Case Study

**Paweł Krause** [1,*] **, Artur Nowoświat** [1] **and Krzysztof Pawłowski** [2]

1    Faculty of Civil Engineering, Silesian University of Technology, 44-100 Gliwice, Poland; artur.nowoswiat@polsl.pl
2    Faculty of Civil and Environmental Engineering and Architecture, UTP University of Science and Technology, 85-796 Bydgoszcz, Poland; krzypaw@utp.edu.pl
*    Correspondence: pawel.krause@polsl.pl

**Abstract:** This paper presents a case study on how to improve the energy efficiency of an institutional building of significant heritage value through retrofitting the external wall system. This building is located in Upper Silesia, Poland. Due to the architectural value of the facade, thermal insulation had to be applied from the inside. As part of this publication, basing on the measurements and simulations, the authors present the results involving the improvement of energy efficiency of the insulated wall. On this basis, they also demonstrate the impact of insulation from the inside on the change of humidity inside the room. The tests were carried out both quantitatively by means of heat flux measurement and qualitatively by means of infrared temperature measurement. The research was supported by numerical modeling. The obtained results indicate that the thermal insulation used in the form of mineral insulation boards applied from the inside improves thermal insulation of the wall. Thus, heat losses through the examined envelope were limited. Computer simulations indicated that no condensation may occur under the condition considered.

**Keywords:** Internal insulation; thermal imaging; heat flux measurement; thermal transmittance

## 1. Introduction

The reduction of energy consumption is currently one of the most important political challenges in the world [1]. Such an approach is also supported by the policy propagated by the European Union [2,3]. The most common method used to improve thermal insulation of buildings is to insulate the walls from the outside [4]. For this purpose, we can consider recycled materials [5,6] and other modifications of insulation materials [7–9]. Polystyrene with the addition of graphite is the most commonly applied modified material [10]. Thermal performance of buildings can be determined by temperature measurements [11], and in such cases, the impact of connectors with metal pins [12] cannot be ignored. The said connectors have much higher heat conduction than the structural wall or the insulation, and therefore they have impact on the rise of heat loss of the building [6].

In engineering practice, there are also articles describing methods for the design of internal insulation [13,14]. This type of thermal insulation is used in historical buildings [15,16]. Thermal insulation applied from the inside significantly affects the change of such parameters as:

- humidity parameters such as condensation [17] or mold growth [15],
- acoustic parameters such as reverberation time [18],
- heat accumulation capacity [19,20].

Despite many problems associated with it, the said technique may be the only way to reduce heat loss in historical buildings, in which any type of applied thermal improvements should not disrupt the

special function of such buildings [15]. Literature offers assessment methods for internal insulation when best solutions for renovated buildings are deliberated [21]. Such assessments are carried out by means of experiments and numerical modeling [22].

In order to be able to use such insulation from the inside, the problem of interstitial or surface condensation must be resolved. Such problems may arise principally in local climates with low temperatures in autumn and winter. Frequently, numerical methods are used to assess the risk of condensation when designing the right thickness and type of insulation [17]. The effectiveness of such solutions depends on the type of applied insulation which should have better moisture transport properties. This, in turn, should contribute to fast redistribution of moisture, reducing the risk of moisture condensation [23]. The materials applied in thermal insulation solutions can be very different, comprising such materials as OSB [24], plasters [25], or paints [26]. Additionally, various studies have demonstrated experimentally or by simulation that a vapor-tight barrier is needed on the inside of the thermal insulation layer to avoid surface condensation during low temperatures [27]. However, for internally insulated masonry walls, such a system may make the walls dry out in summer [28]. Therefore, when designing such thermal insulation, we should take into account a tight or vapor-tight internal insulation system, external and internal environmental conditions, and wall structure [29].

Unfortunately, when insulating from the inside, apart from the risk of condensation, there are also other problems [30] related primarily to thermal bridges that are difficult to avoid [31,32]. Such problems can comprise mold development, biological corrosion, the decrease of thermal resistance, and increased heat loss.

Besides, the research on thermal bridges is a very important issue when investigating energy efficiency of buildings [33]. In the calculations, thermal bridges are very often included as an additional element to the calculated heat transfer coefficient [34]. In the studies on thermal bridges and their influence on heat losses in a building, a two-dimensional FEM model in a steady state is often used to model structural nodes [35]. The assessment process of energy efficiency of well-insulated buildings very often neglects thermal bridges, and after all they can play a role in heat loss through a building envelope [36].

The identification of the imperfections of building envelopes displayed by increased heat losses is often performed using in situ measurements. One of the measurement options is the infrared method [37–40].

The present article analyzes one specific implementation of thermal insulation from the inside of a wall with an untypical external facade. For this purpose, mineral insulation boards were used as insulation material.

The analysis of in situ measurements was also compared with computer simulations used to estimate the risk of condensation. The studies demonstrated better insulation properties of the wall in effect of the application of thermal insulation.

## 2. Methodology

### 2.1. Subject of Research

This article focuses on the method of assessing thermal insulation of a wall from the inside. The assessment can be made through two steps as follows:

I.　　Before making the insulation:

　　Assessment on the basis of the measurement method of thermal transmittance,
　　Assessment of temperature distribution using the infrared measurement method,
　　Assessment of thermal bridges—FEM modeling (assumptions as for the infrared measurement).

II.　　After making the insulation:

　　Everything as in point I.

The study involved an external wall in an educational building of a University located in the Region of Upper Silesia (Poland). The part of the building above ground level was made as prefabricated reinforced concrete frame structure, namely SF-64, which is consisted of:

- external reinforced concrete frames "Z4",
- internal prefabricated units "W",
- ceilings made of hollow core slabs "S".

The outer longitudinal walls are made of prefabricated reinforced concrete blocks with a corrugated outer surface mounted on transoms. Gable walls, in compliance with the archival design documentation from the 1960s, were made in the form of foam-gas-silicate blocks (Table 1). The façade is finished with ceramic façade tiles of the dimensions of 0.185 m × 0.09 m and the thickness of 0.02 to 0.03 m. The texture of the façade in window sill strips and attics was concrete and corrugated. On one of the outer walls of the building there are glazed ceramic tiles which are viewed as an element of "contemporary art" (Figure 1).

**Table 1.** Summary of thermal resistance for the gable wall (with mosaic)—the existing state.

| Type of Material | Thermal Conductivity λ [W/mK] | Thickness d [m] | Resistance R [m$^2$K/W] |
|---|---|---|---|
| Inner side of the envelope | - | | 0.13 |
| Cement-lime plaster | 0.820 | 0.015 | 0.018 |
| Block PGS 800 (wet conditions) | 0.480 | 0.240 | 0.500 |
| Ceramic tiles | 1.050 | 0.030 | 0.029 |
| Outer side of the envelope | | | 0.04 |
| **Total** | | | **0.717** |

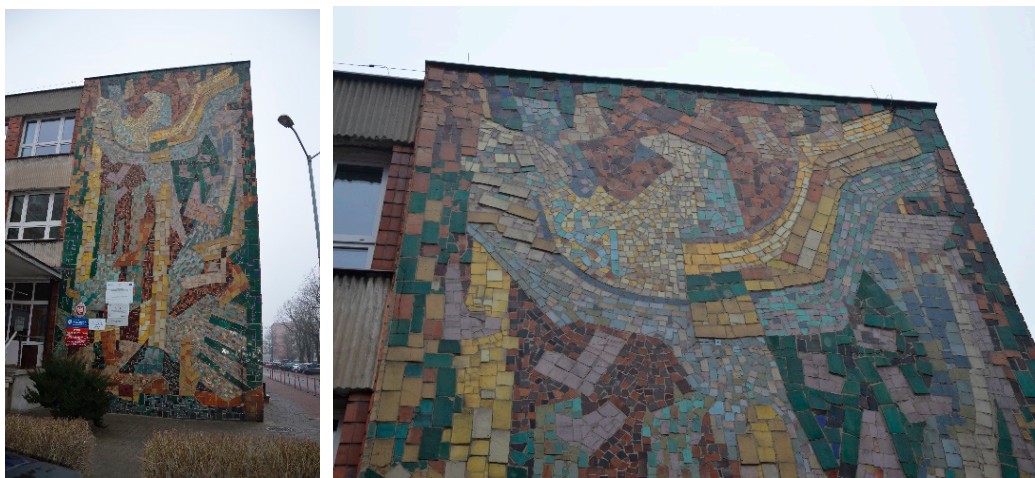

**Figure 1.** View of a fragment of the facade with glazed ceramic tiles.

Due to the ceramic tiles on the facade, it was impossible to apply thermal insulation on the outer surface of the wall. Therefore, thermal insulation was used on the inner surface of that wall.

The entire thermal modernization process consisted in partial thermal insulation of the walls from the outside and the insulation of the gable wall from the inside. For the internal insulation, mineral insulation boards with a thickness of 0.06 m were applied, made of light cellular concrete with a density of 115 kg/m$^3$. The boards of the dimensions of 0.6 m × 0.39 m were installed on adhesive mortar and additionally fixed with mechanical connectors made of PVC plastic.

The building after the thermal modernization process is presented in Figure 2a. In Figure 2b, we can see the projection of a fragment of the building with the marked gable wall.

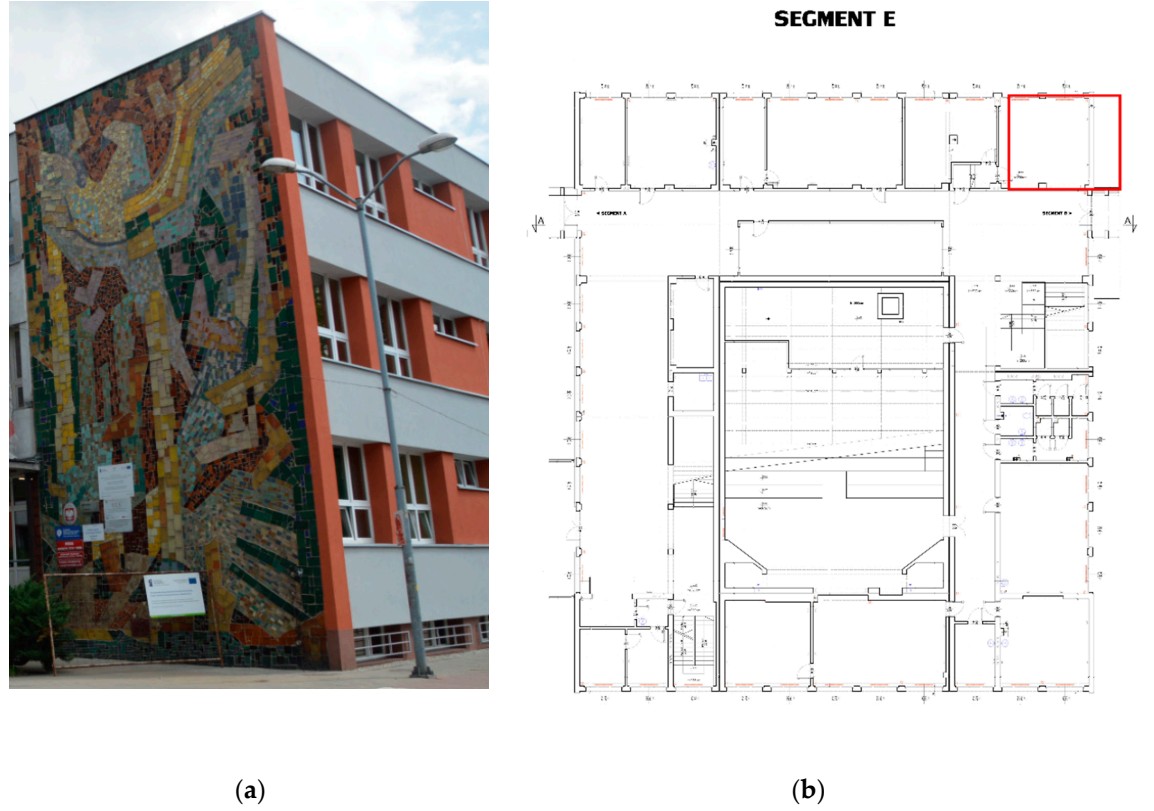

(**a**)　　　　　　　　　　　　　　　　　　　　　(**b**)

**Figure 2.** (**a**) Fragment of the building facade insulated partly from the outside and partly from the inside; (**b**) projection of a fragment of the building with the marked gable wall.

Figure 3 presents the classroom in which the wall was insulated.

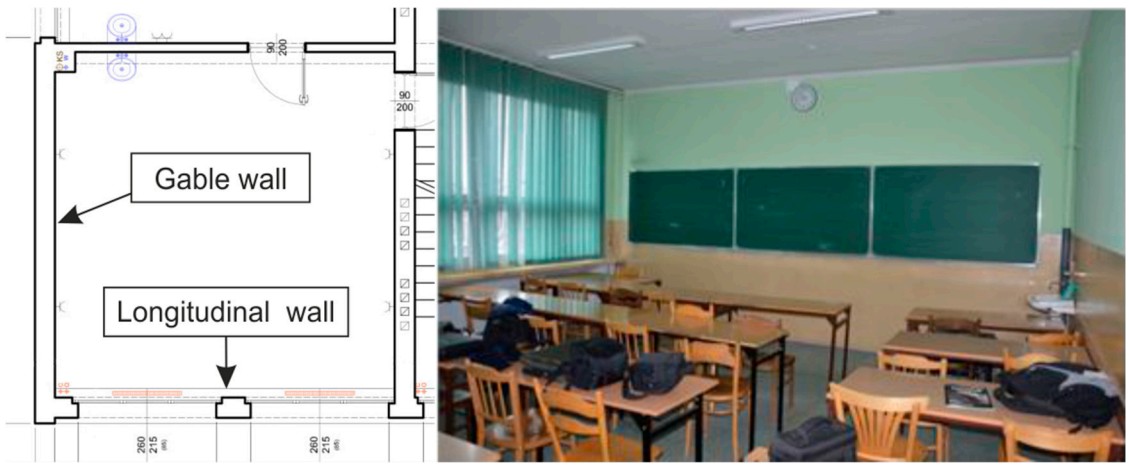

**Figure 3.** Projection and view of the room adjacent to the gable wall.

In order to fully illustrate the thermal state of the building, basing on the design represented by the projection in Figure 3, calculations were carried out, as presented in Tables 1–3.

**Table 2.** Summary of thermal resistance for the longitudinal wall—the existing state.

| Type of Material | Thermal Conductivity λ [W/mK] | Thickness d [m] | Resistance R [m²K/W] |
|---|---|---|---|
| Inner side of the envelope | | | 0.13 |
| Cement-lime plaster | 0.820 | 0.015 | 0.020 |
| Prefabricated wall | 2.300 | 0.350 | 0.150 |
| Outer side of the envelope | | | 0.04 |
| **Total** | | | **0.340** |

**Table 3.** Summary of thermal resistance for the gable wall (with mosaic)—after insulation.

| Type of Material | Thermal Conductivity λ [W/mK] | Thickness d [m] | Resistance R [m²K/W] |
|---|---|---|---|
| Inner side of the envelope | - | | 0.13 |
| Gypsum plaster finish | 0.55 | 0.005 | 0.009 |
| Mineral insulation boards | 0.043 | 0.060 | 1.395 |
| Cement-lime plaster | 0.820 | 0.015 | 0.018 |
| Block PGS 800 (wet conditions) | 0.480 | 0.240 | 0.500 |
| Ceramic tiles | 1.050 | 0.030 | 0.029 |
| Outer side of the envelope | | | 0.04 |
| **Total** | | | **2.121** |

The calculated thermal transmittance based on ISO 6946 [41] is:

$$U = 1.395 \frac{W}{m^2 K}$$

The calculated thermal transmittance based on ISO 6946 [41] is:

$$U = 2.937 \frac{W}{m^2 K}$$

The calculated thermal transmittance based on ISO 6946 [41] is:

$$U = 0.471 \frac{W}{m^2 K}$$

In order to assess the thermal condition of the investigated wall, temperature measurements were made with the use of a video camera. The simulations only involved the determination of temperatures at specific places of the envelope. The said tests were complemented by the measurements of thermal resistance. The case study involved quick in situ measurements and quick numerical verification, ignoring the theoretical issues of heat and mass transport, which could be analyzed using the Fick's or Fourier's equations.

### 2.2. Measurements Using a Thermal Imaging Camera

The measurement with a thermal imaging camera was made to identify the state of thermal protection of the external wall. The tests were carried out both from the outside and from the inside. For testing in real conditions, a thermal imaging camera was applied with the temperature measurement range of −20 °C ÷ +1500 °C, resolution of 161,472 pixels, thermal sensitivity for the lens 42° × 32° <30 mK, IFOV spatial resolution for the lens 42° × 32° of 2.41 mrad/pixel and interval spectral sensitivity of 7.5–14 μm. In addition, a temperature logger was applied with the measuring range of −40 °C ÷ +70 °C and accuracy of ± 0.2 °C, as well as a humidity logger with the range of 0–100% and accuracy of ±2%.

The tests comprised the registration of temperature distribution on wall surfaces from the inside and outside. As part of the measurements, temperatures at selected characteristic points

were determined, and the occurring thermal bridges were defined. The measurements outside were conditioned by the absence of rainfall and the reduction of wind speed to $v < 1$ m/s.

### 2.3. Measurements of Heat Flux

The measurement of heat flux was made using a system consisting of the gSKIN DLOG-4231 measuring and logging device with a set of measuring sensors. The gSKIN-XO 67 7C heat meter of the dimensions of 30 mm × 30 mm × 2.5 mm is based on a thermoelectric detector with the measuring surface material made of silicon. The measurement range of the logger is ± 3.13 mV, resolution of 1.6 µV, and the equivalent noise of heat flux per unit area is 0.073 W/m$^2$. The resolution of heat flux with reference to the surface is 0.09 W/m$^2$.

Knowing the values of temperature difference on both sides of the envelope and the heat flux value, the total thermal resistance of the wall was determined by simply adding the value of heat flux density and the temperature difference of the media on both sides of the envelope using the formula (1). Obviously, it is an averaged method in compliance with ISO 9869 [42].

$$R = \frac{\sum\limits_{j=1}^{n} \left(t_{ij} - t_{ej}\right)}{\sum\limits_{j=1}^{n} q_j} \tag{1}$$

where:

$t_i$—ambient temperature inside the building, °C
$t_e$—ambient temperature outside the building, °C
$q$—heat flow rate, $\frac{W}{m^2}$
$j$—time interval

In line with the measurement procedure and measurement experience [43,44] the following assumptions should be used:

The measurement period should take at least 72 h with a specific range of sampling and logging intervals.

1.  The R-value obtained from the last two measurement days should not differ by more than 5% [42].
2.  The difference between R-values obtained from the first and last certain number of days is within 5% [42].

## 3. Results and Discussion

### 3.1. Measurements of Temperature and Heat Flux

The emissivity of finishing layers of the wall was determined by the comparative method with respect to the reference tape with the known emissivity of 0.96. The emissivity of the inner plaster determined on this basis is 0.92. Due to differences in the emissivity of the individual selected types of façade ceramic tiles, ranging within 0.89–0.93, the average emissivity of 0.91 was determined for a given surface. The temperature of indoor and outdoor air for the measurement No. 1 (the wall without thermal insulation) was respectively $t_i = 19.3$ °C, $t_e = 2.7$ °C, while for the measurement No. 2 (after applying the insulation from the inside) it was $t_i = 17.0$ °C, $t_e = -2.5$ °C. The results of thermal imaging measurements before and after the application of the insulation layer on the inner wall (Figure 2) surface are presented in Figure 4, and the results of temperature as a function of distance px are presented in Figures 5–7.

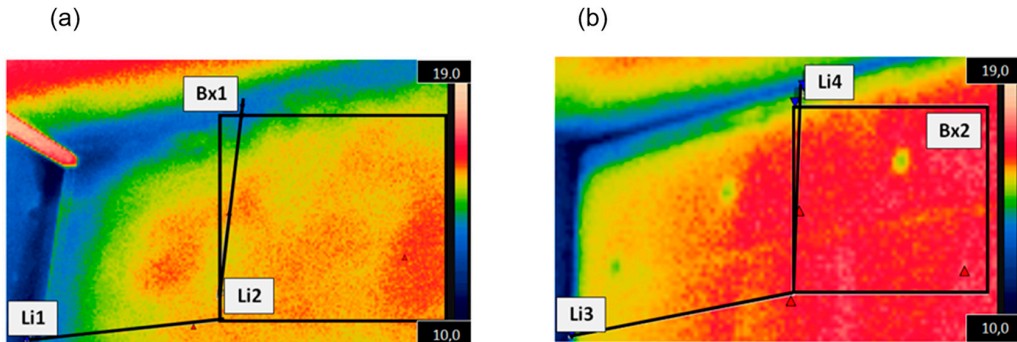

**Figure 4.** Temperature distribution in color scale on the wall surface from the side of heat inflow (**a**) without thermal insulation (**b**) with thermal insulation from the inside.

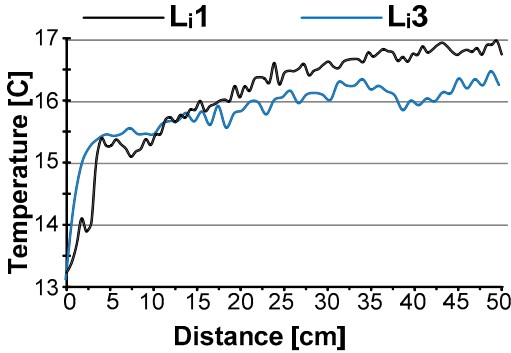

**Figure 5.** Temperature distribution along the lines Li1 and Li3.

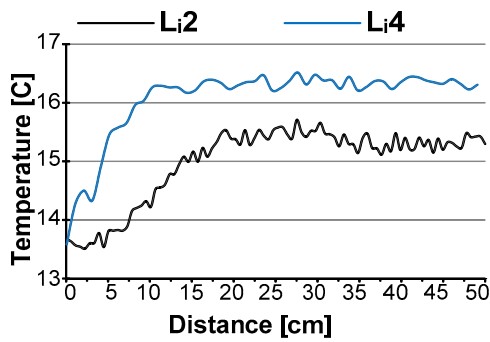

**Figure 6.** Temperature distribution along the lines Li2 and Li4.

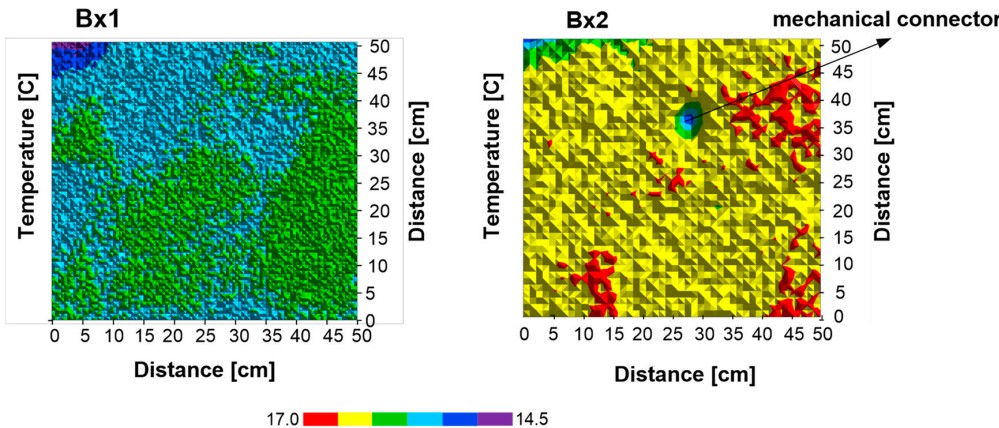

**Figure 7.** Temperature distribution on the surfaces Bx1 and Bx2.

The analysis of thermographic measurement results before the application of the insulation (Figure 4a) and after the application of the insulation (Figure 4b) for different environmental conditions on both sides of the envelope indicates that there is a change in the character of temperature distribution. It is demonstrated on the diagrams along the lines Li3 and Li4 (Figures 5 and 6). The course of temperature changes along the line Li3 determined from the corner of the outer walls shows that there is a significant reduction of the impact of the geometric linear bridge after the insulation. We can observe a sharp change in temperature on the outer wall surface at the distance not exceeding 0.15 m, followed by its stabilization. Before the insulation, the temperature on the wall surface along the line Li1 stabilized at the distance of about 0.55 m from the corner (Figures 5 and 6). A similar situation can be observed for the line Li2 at the junction between the wall and the ceiling (Figures 5 and 6). The measurements were made at a small angle to the wall surface in order to minimize the influence of reflected radiation. If the measurements had been made at right angles to the surface, they could have been distorted. Based on the information about the distance of the thermal imaging camera from the tested wall and knowing the selected actual characteristic dimensions, lines Li1 and Li3 perpendicular to the curtain wall of the same direction and approximately the same length were determined. Similarly, lines Li2 and Li4 perpendicular to the ceiling surface were determined. When examining the temperature field Bx1 before the thermal insulation, we can observe its slight surface variation (Figure 7). The lowest temperature was recorded in the fragment of the wall-to-ceiling connection. The distribution of temperature on the insulated surface (Bx2) allows to identify connections between individual thermal insulation elements in which a slight decrease in temperature on the inner wall surface is observable (Figure 4). Additionally, a decrease in temperature on the surface of the insulation is observed at the place of the existing mechanical connector (Figure 7). This may be caused by the change of heat conductivity coefficient, although a slight influence of the change in emissivity at this place cannot be ruled out.

The heat flux sensor was placed from the inside at the height of 1.0 m above floor level, at a distance of 2.0 m from the outer wall. In order to ensure good quality of the carried out measurements, proper thermal contact of the connection between the heat meter and the outer wall surface was ensured by the use of conductive adhesive contact paste. Parallel to the measurement of heat flux density, air temperature was measured on both sides of the wall using temperature sensors contained in the measurement setup. The results were recorded in hourly time steps. The second measurement was taken at the same place after the wall had been insulated from the inside.

The carried out measurements demonstrated a significant difference in the value of thermal transmittance of the outer wall in relation to the calculation results (Table 4). The calculations based on the data contained in the archival project documentation carried out in line with the methodology ISO 6946: 2017 [41] demonstrated that thermal transmittance of the wall in question was $U = 1.40$ W/m$^2$K. The tests carried out before wall insulation allowed us to determine the actual value of thermal transmittance. The measurement with the use of a heat flux sensor showed that thermal transmittance was $U = 2.37$ W/m$^2$K. The tests carried out after the application of thermal insulation, which was 0.06 m thick and had thermal conductivity coefficient of $\lambda = 0.042$ W/mK, demonstrated that thermal transmittance decreased to the level of $U = 0.56$ W/m$^2$K. This value is higher by 0.09 W/m$^2$K as compared to the calculated thermal transmittance (Tables 1 and 2).

The heating system in the room was made with the use of panel radiators. The heat source was located in the local gas boiler room. Thermographic measurements were taken during the semester break, when the temperature of the internal air decreased. The first measurement was made at the beginning of this period, for the temperature of +19.3 °C. The outside air temperature was 2.7 °C. The next measurement after the completion of thermal insulation works was carried out at a lowered internal air temperature of 17.0 °C and lowered external air temperature of −2.5 °C.

The use of thermal insulation from the inside of the wall also has a significant impact on the microclimate in the room. The measurement results of the indoor air temperature, humidity, and dew point temperature are presented in Figure 8.

**Table 4.** Calculated and measured thermal transmittance.

| Characteristics | Symbol | Calculations Based on the Archival Project | Measurement without Thermal Insulation | Measurement with Thermal Insulation |
|---|---|---|---|---|
| Duration of measurements [h] | $\tau_{total}$ | | 118 | 110 |
| Analyzed time period [h] | $\tau_{analysis}$ | | 96 | 96 |
| Average heat flux [W/m$^2$] | $q_{javerage}$ | | 26.3 | 14.8 |
| Thermal transmittance [W/(m$^2$K)] | $U$ | 1.40 | 2.37 | 0.56 |
| Thermal resistance [(m$^2$K)/W] | $R$ | | 0.42 | 1.79 |
| Theoretical thermal resistance (Tables 1 and 3) | $R$ | | 0.717 | 2.121 |

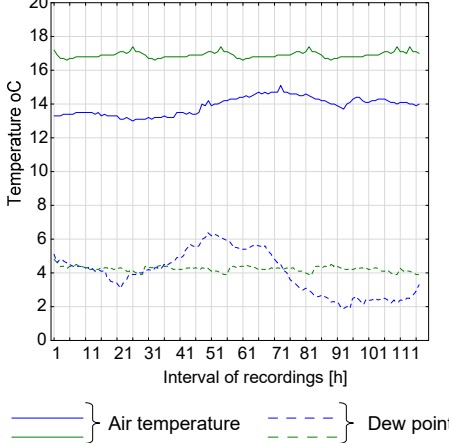

**Figure 8.** Measurement of indoor microclimate. Blue color—the measurement without thermal insulation taken on 27 January 2018–01 February 2018 Green color—the measurement with thermal insulation taken on 17 February 2018–18 February 2018.

The measurement frequency was one measurement per hour. Stabilized results were accepted for the presentation of results. The diagrams demonstrate principally a significant change of indoor air humidity. After the application of internal insulation, the relative humidity measured around the corner of the outer walls definitely decreased. It is related to the moisture level of wall substrate. By the use of a thermal insulation layer with high water vapor permeability, the excess water vapor contained in the air was absorbed by the thermal insulation layer. The results are collected in Tables 5 and 6.

**Table 5.** Indoor microclimate before the application of insulation. Measurement taken from 27 January 2018 to 01 February 2018.

| | Air Temperature [°C] | Air Humidity [%] | Dew Point [°C] |
|---|---|---|---|
| **Min** | 13.0 | 43.6 | 3.9 |
| **Max** | 15.1 | 60.0 | 4.8 |
| **Average** | 13.9 | 51.6 | 4.0 |

**Table 6.** Indoor microclimate after the application of insulation. Measurement taken from 17 February 2018 to 18 February 2018.

| | Air Temperature [°C] | Air Humidity [%] | Dew Point [°C] |
|---|---|---|---|
| **Min** | 16.6 | 41.5 | 3.0 |
| **Max** | 17.4 | 44.3 | 4.2 |
| **Average** | 16.9 | 43.2 | 4.3 |

### 3.2. Simulations

As a preliminary assumption, we accepted that the simulations would be carried out in the same climatic conditions as during the measurement. Owing to this, the simulation complements the in-situ measurement, which together allows for quick thermal diagnostics of the wall.

The values of linear thermal transmittance $\Psi$ and point thermal transmittance $\chi$ depend on the configuration and thermal conductivity of material layers and on how the calculations of one-dimensional thermal transmittance were made, i.e., which values of coefficients $U$ were selected for different building elements and which planes they correspond to. In most cases, heat loss calculations can be performed using the 2D model [45]. Heat losses can be expressed by the linear heat flux Q, [W/m]. The total heat flux of 2D, taking into account the presence of a thermal bridge, is determined from the relationship:

$$q_l = \left( \sum_i U_i \cdot l_i + \Psi \right) \cdot \Delta t_{i,e} \quad \left[ \frac{W}{m} \right] \tag{2}$$

or

$$q_l = L^{2D} \cdot \Delta t_{i,e} \quad \left[ \frac{W}{m} \right] \tag{3}$$

where:

$q$—total heat flux 2D, taking into account the presence of a thermal bridge

$U$—thermal transmittance of the envelope in the one-dimensional field 1D

$L$—length

$\Psi$—linear thermal transmittance resulting from the presence of the bridge 2D

$\Delta t_{i,e}$—temperature difference

$L^{2D}$—linear coupling factor between two environments, [W/(m·K)] and $L^{2D}$ is the term on the left on the temperature difference in Equation (2).

In the calculations related to heat losses, the coefficients related to the occurrence of linear thermal bridges are applied. These include linear thermal transmittance coefficient in relation to the part of the wall-to-window connection $\Psi_{iO}$ and the linear thermal transmittance coefficient with regard to the connection of the internal and external wall $\Psi_{iwall}$.

With respect to the connection of the outer wall with the ceiling in the cross-section through the ceiling ring, we determined the branch thermal transmittance for the lower part of the connector $\Psi_{id}$ and for the upper part of the connector $\Psi_{ig}$. The procedure for calculating branch thermal transmittance coefficients $\Psi$ consists in:

- separation of internal branches of the thermal bridge and assigning initial and boundary conditions,
- calculation (numerically) of heat fluxes flowing through the separated branches (parts) of the bridge,
- calculation of the relevant branch coefficients in line with respective relations, using the data corresponding to the separated branches.

The value of the temperature factor $f_{Rsi(2D)}$ was determined in compliance with the relationship:

$$f_{Rsi(2D)} = \frac{t_{\min} - t_e}{t_i - t_e} \tag{4}$$

where:

$t_{min}$—minimum temperature on the internal surface of the envelope at the place of thermal bridge [°C],

$t_e$—temperature of outdoor air [°C],

$t_i$—temperature of indoor air [°C].

The calculations presented in the article were performed with the use of computer programs: TRISCO-KOBRU 86, used for thermal analysis of stationary heat flow (under constant ambient temperature conditions), under various boundary conditions. The geometry was described by a list of

regular blocks of points on a regular mesh. In the program, materials and boundary conditions with different thermal properties are marked with different colors. Each block consists of a specific material and has specific boundary conditions.

A finite element mesh with a uniform side equal to 1 cm (in some cases even 0.5 cm) was adopted, in compliance with the guidelines of ISO 10211: 2008 [46]. In the applied version of the program TRISCO-KOBRU 86, it is possible to define a constant temperature between the edges of specific blocks. This tool allows to carry out calculations with reference to building connectors in the (2D) layout.

After the creation of geometry and the adoption of boundary conditions, the calculation process is started. The temperature field is calculated using the matrix of linear equations. The carried out calculations provide a graphical and digital result comprising temperatures and heat fluxes of the analyzed building connector.

According to the instructions of the program, the calculation parameters were as follows:

- the maximum number of iterations—10,000,
- the absolute error in the calculation of temperatures—0.0001 °C,
- the absolute error in the calculation of heat fluxes in the connector—0.001%.

In the assessment of thermal bridges, the choice of boundary conditions is of key importance, especially as they may differ depending on the type of calculation—Table 7.

**Table 7.** Determination of boundary conditions for the calculation of thermal bridges—own study based on ISO 6946 and ISO 13877.

| Parameter | Objective of Calculations | |
|---|---|---|
| | Risk Assessment Involving the Development of Mold and Mold Fungi acc. ISO 13788:2012 | Calculations of Heat Flux acc. ISO 6946:2017 |
| Temperature of outdoor air ($t_e$) and indoor air ($t_i$) [°C] | $t_e$—based on the division of Poland into climatic zones $t_i$—based on the Regulation "technical conditions" | $t_e$—based on the division of Poland into climatic zones $t_i$—based on the Regulation "technical conditions" |
| Thermal transfer resistance on the internal ($R_{se}$) and external ($R_{si}$) surface of the envelope [(m²·K)/W] | $R_{si} = 0.13$ for glazing and frames, $h_i = 1/R_{si} = 7.69$ $R_{si} = 0.25$ for all other elements, $h_i = 1/R_{si} = 4.0$ $R_{se} = 0.04, h_e = 1/R_{se} = 25$ | $R_{si} = 0.10$ (upward flow), $h_i = 1/R_{si} = 10$ $R_{si} = 0.13$ (horizontal flow), $h_i = 1/R_{si} = 7.69$ $R_{si} = 0.17$ (downward flow), $h_i = 1/R_{si} = 5.88$ $R_{se} = 0.04, h_e = 1/R_{se} = 25$ |

The Figure 9 presents an example situation with the mesh and boundary conditions. The same boundary conditions are adopted later in the work.

The calculation process of thermal bridges with the use of a computer program consists of several stages:

- modeling of connectors—thermal bridges,
- adoption of boundary conditions,
- defining the material characteristics of thermal bridges,
- calculations of hygrothermal parameters of thermal bridges,
- developing a catalog of thermal bridges.

Modeling, i.e., the division into several geometric models, must be done by selecting appropriate division planes in compliance with the standard ISO 10211: 2017 [46].

It should also be noted that at the junction of the outer wall with the window on the inner surface, the temperature drops to the level of $t_{min} = 11.02$ °C, which brings about the risk of the development of surface condensation (the risk of the development of mold and mildew fungi).

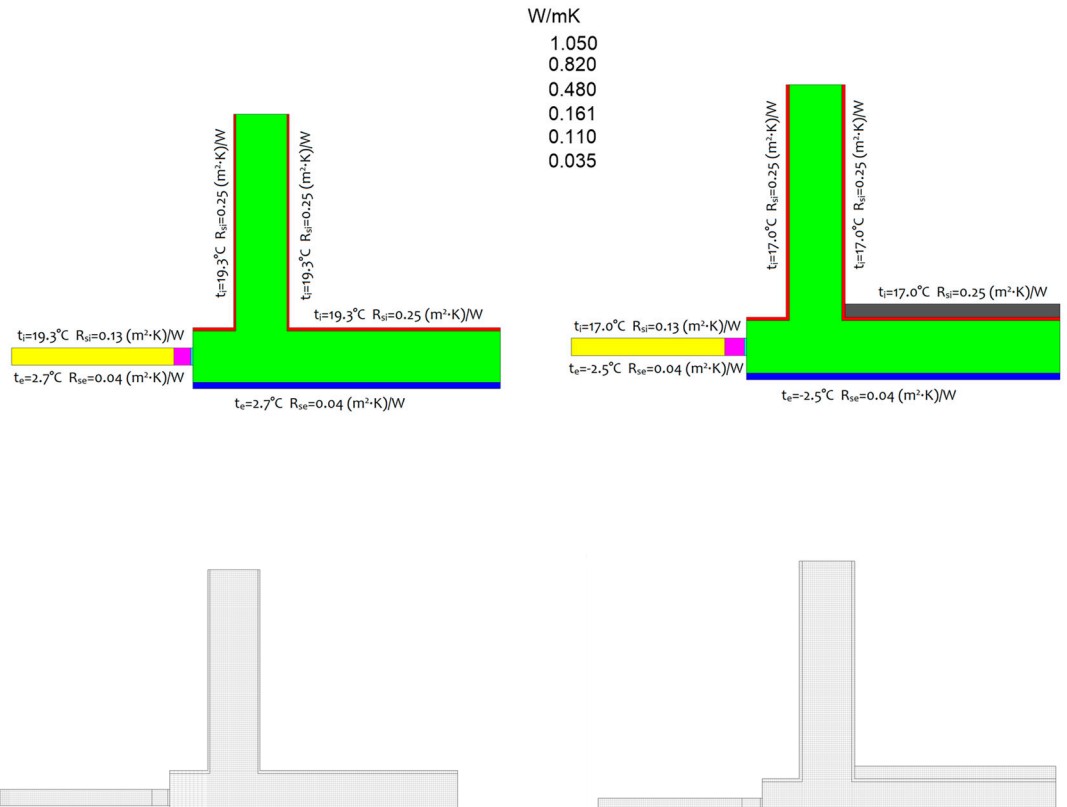

**Figure 9.** Boundary conditions and mesh compaction used in the modeling.

The first series of simulation results involved the connection of the outer wall with the inner wall are in Figure 10.

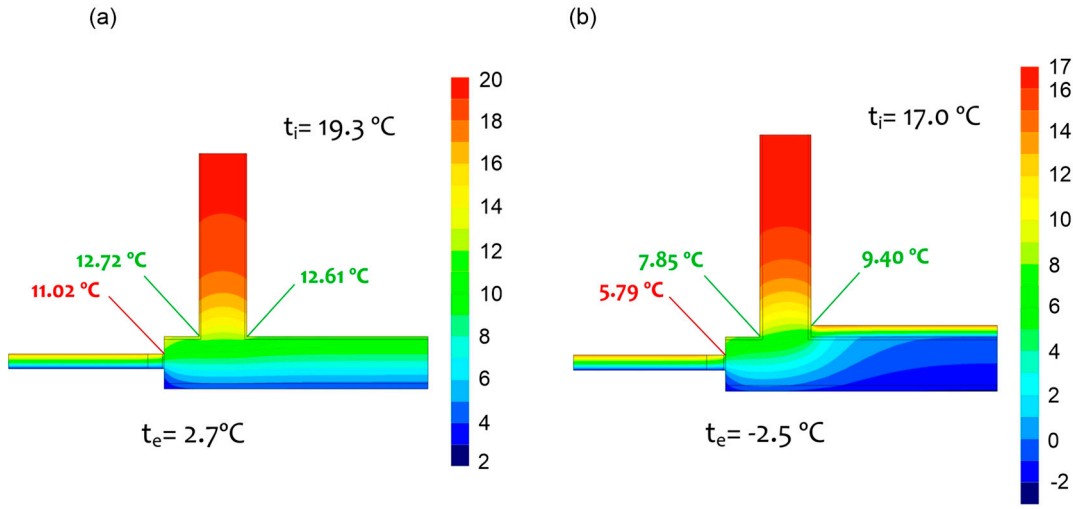

**Figure 10.** Simulated temperature distribution of the connection of outer and inner walls: (**a**) simulations without thermal insulation; (**b**) simulations with thermal insulation.

The simulation results involving physical parameters are presented in Table 8, where: $U_s$—thermal transmittance of the wall, $U_f$—thermal transmittance of the window frame, $U_g$—thermal transmittance of glazing.

**Table 8.** Physical parameters obtained by the simulation.

| | $U_c$ [W/(m²·K)] | $\Phi$ [W] | $L^{2D}$ [W/(m·K)] | $\Psi_i$ [W/(m·K)] | $t_{minimal}$ [°C] | $f_{Rsi(2D)}$ [-] |
|---|---|---|---|---|---|---|
| | **Physical Parameters** | | | | | |
| | **before thermal insulation** | | | | | |
| (a) | $U_s = 2.37$ $U_f = 1.50$ $U_g = 1.10$ | 75.58 $\Phi_s = 58.34$ $\Phi_o = 17.24$ | 4.55 $L_s^{2D} = 3.51$ $L_o^{2D} = 1.04$ | 0.78 $\Psi_{i,s} = 0.71$ $\Psi_{i,o} = 0.07$ | 11.02 | 0.501 |
| | **after thermal simulation** | | | | | |
| (b) | $U_{s1} = 0.56$ $U_{s2} = 2.37$ $U_f = 1.50$ $U_g = 1.10$ | 59.20 $\Phi_s = 41.11$ $\Phi_o = 18.09$ | 3.04 $L_s^{2D} = 2.11$ $L_o^{2D} = 0.93$ | 1.13 $\Psi_{i,s} = 1.125$ $\Psi_{i,o} = 0.005$ | 5.79 | 0.430 |

The next analysis involved the connection of outer walls as in Figure 11.

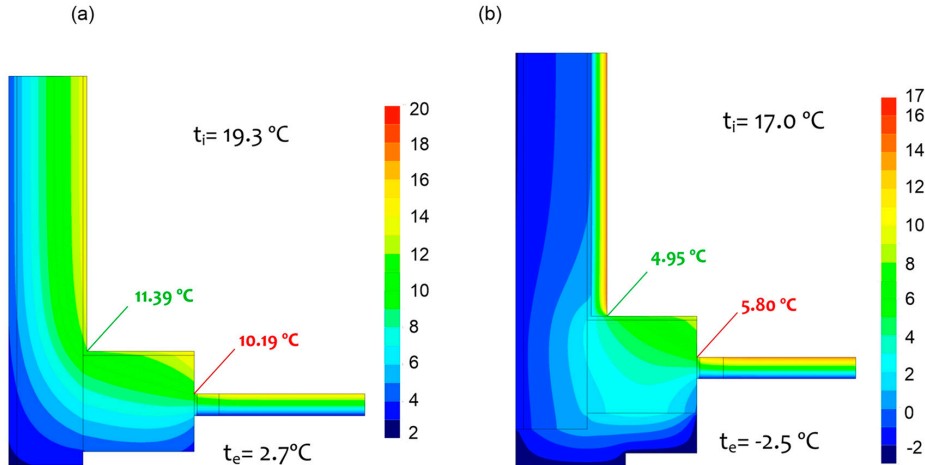

**Figure 11.** Simulated temperature distribution, of the connection of outer walls: (**a**) simulations without thermal insulation; (**b**) simulations with thermal insulation.

The simulation results involving physical parameters are presented in Table 9.

**Table 9.** Simulations involving physical parameters obtained by the simulation.

| | $U_c$ [W/(m²·K)] | $\Phi$ [W] | $L^{2D}$ [W/(m·K)] | $\Psi_i$ [W/(m·K)] | $t_{min}$ [°C] | $f_{Rsi(2D)}$ [-] |
|---|---|---|---|---|---|---|
| | **Physical Parameters** | | | | | |
| | **before thermal insulation** | | | | | |
| (a) | $U_{s1} = 2.37$ $U_{s2} = 2.937$ $U_f = 1.50$ $U_g = 1.10$ | 80.30 $\Phi_s = 66.92$ $\Phi_o = 13.38$ | 4.84 $L_s^{2D} = 4.03$ $L_o^{2D} = 0.81$ | 0.620 $\Psi_{i,s} = 0.515$ $\Psi_{i,o} = 0.105$ | 10.19 | 0.45 |
| | **after thermal insulation** | | | | | |
| (b) | $U_{s1} = 0.56$ $U_{s2} = 0.193$ $U_f = 1.50$ $U_g = 1.10$ | 50.95 $\Phi_s = 35.86$ $\Phi_o = 15.10$ | 2.61 $L_s^{2D} = 1.84$ $L_o^{2D} = 0.77$ | 1.29 $\Psi_{i,s} = 1.22$ $\Psi_{i,o} = 0.07$ | 5.80 | 0.430 |

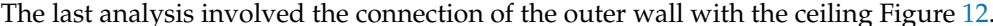

The last analysis involved the connection of the outer wall with the ceiling Figure 12.

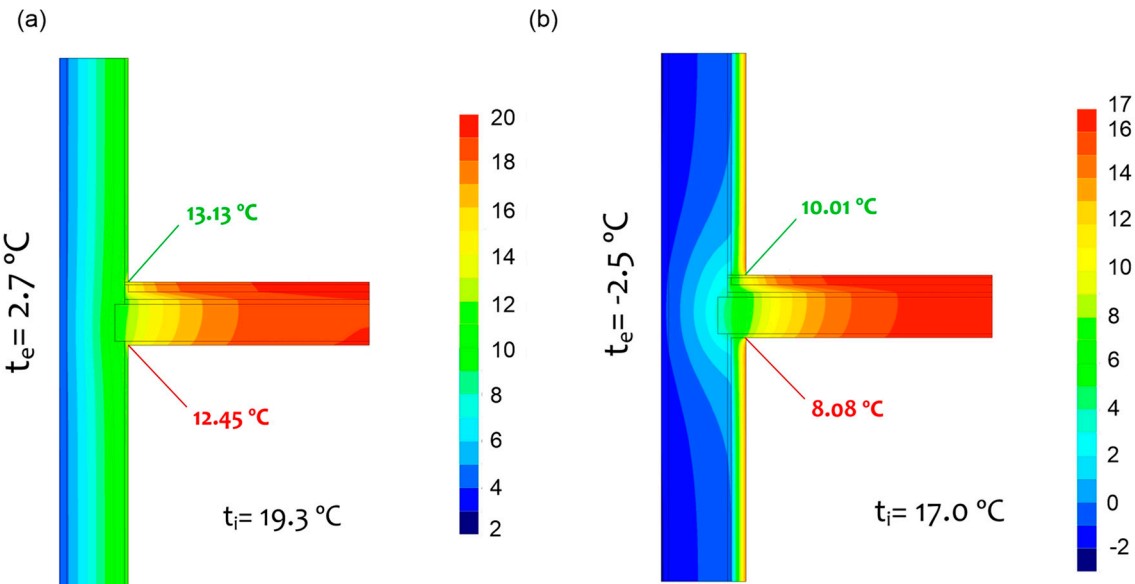

**Figure 12.** Simulated temperature distribution of the connection of outer walls: (**a**) simulations without thermal insulation; (**b**) simulations with thermal insulation.

The simulation results for physical parameters are presented in Table 10.

**Table 10.** Physical parameters obtained by the simulation.

| | $U_c$ [W/(m²·K)] | $\Phi$ [W] | $L^{2D}$ [W/(m·K)] | $\Psi_i$ [W/(m·K)] | $t_{min}$ [°C] | $f_{Rsi(2D)}$ [-] |
|---|---|---|---|---|---|---|
| | | | **Physical Parameters** | | | |
| | | | **before thermal simulation** | | | |
| (a) | $U_s = 2.37$ $U_s = 2.37$ | 85.14 $\Phi_d = 46.69$ $\Phi_g = 38.45$ | 5.13 $L_d{}^{2D} = 2.81$ $L_g{}^{2D} = 2.31$ | 0.603 $\Psi_{i,d} = 0.44$ $\Psi_{i,g} = 0.163$ | 12.45 | 0.587 |
| | | | **after thermal simulation** | | | |
| (b) | $U_s = 0.56$ $U_s = 0.56$ | 35.71 $\Phi_d = 22.04$ $\Phi_g = 13.67$ | 1.83 $L_d{}^{2D} = 1.13$ $L_g{}^{2D} = 0.70$ | 0.768 $\Psi_{i,d} = 0.578$ $\Psi_{i,g} = 0.190$ | 8.08 | 0.542 |

The connections of the outer wall with the internal wall together with the window generate the following heat losses before the insulation: heat flux $\Phi = 75.58$ W, thermal coupling coefficient $L^{2D} = 4.55$ W/(m·K) and linear thermal transmittance coefficient $\Psi = 0.78$ W/(m·K). It should also be noted that at the juncture of the outer wall with the window on the inner surface the temperature is reduced to the level of $t_{min} = 11.02$ °C—Table 8.

In the second case, one branch of the outer wall was insulated from the inside with cellular concrete blocks of $\lambda = 0.042$ W/(m·K). Therefore, the obtained value of thermal transmittance of the outer wall (after insulation) was $U_{s1} = 0.56$ W/(m²·K). Such an insulation method generates lower values of heat loss $\Phi = 59.20$ W, $L^{2D} = 3.04$ W/(m·K) than for the case without insulation. It should be noted that such a connection generates additional heat loss with linear thermal transmittance $\Psi = 1.13$ W/(m·K)—i.e., the value higher than that in the initial variant without insulation. This is due to the calculation method of the coefficient $\Psi$. On the inner surface of the envelope at the connection

point of the outer wall with the window, the minimum temperature is $t_{min}$ = 5.79 °C, which brings about the risk of surface condensation (the risk of mold and mildew fungi development)—Table 8.

The connections of outer walls in the corner with the window before the insulation generate higher heat losses involving heat flux $\Phi$, thermal coupling coefficient $L^{2D}$ than in the case of insulation from the inside. However, as in the first calculation example, additional heat losses regarding linear thermal transmittance are higher $\Psi$ = 1.29 W/(m·K) than in the variant before the insulation $\Psi$ = 0.62 W/(m·K). In addition, the applied insulation brings about a significant reduction of temperature on the inner surface (at the junction outer wall—window) $t_{min}$ = 5.80 °C—Table 9.

And for the connection of the outer wall with the ceiling, the application of insulation from the outside brings about the reduction in heat loss involving heat flux $\Phi$, thermal coupling coefficient $L^{2D}$ and the decrease in temperature on the inner surface of the envelope in the bottom surface of the ceiling $t_{min}$ = 8.08 °C—Table 10.

## 4. Conclusions

Most of the works in world literature that are devoted to thermal insulation from the inside concern historical buildings.

In this article, we do not compare the results of a case study with scientific reports of other authors. Mostly because the results obtained in this article involve special meteorological conditions, an individually selected wall, its connection with other building envelopes, above all, a unique facade. The authors of the present work do not go into the nuances of calculation methods or measurement methods, as e.g., Orlik-Kożdoń in her work [14]. The authors of the work aimed to demonstrate the only way to improve thermal insulation of one wall of the building (the entire building was subjected to thermal modernization process), which was characterized by a very unique and individually selected facade in the form of a mosaic.

Apart from other analyses, the paper demonstrates the effect of the impact of the connections of insulation boards on temperature reduction at these connections, which is presented in Figure 4b, Figure 5, and Figure 6. Furthermore, it was demonstrated that basing on the archival design, the calculated thermal transmittance differs significantly from the measurement results, which is presented in Tables 1 and 4. We can assume that the appropriate parameters were inaccurately assigned to the applied materials and that the parameters of the materials changed during the use of the building. One of the factors having impact on the change in the thermal parameters of the wall may be moisture level increasing over time.

- Based on the carried out simulation calculations, the following conclusions can be drawn: The assessment of the connection in terms of additional heat loss should not be based only on the value of linear thermal transmittance $\Psi$ [W/(m·K)], often used in practice, because in many cases the introduction of additional insulation does not lower its value—all parameters affecting heat losses should be analyzed, i.e.,: $\Phi$ [W], $L^{2D}$ [W/(m·K)], $\Psi$ [W/(m·K)].
- The insulation of envelopes and connections from the inside should be analyzed in terms of lowering the temperature on the internal surface of the envelope, and limiting the risk of surface condensation (the risk of mold and mildew fungi development).
- By determining branch thermal transmittance coefficients for a single part of the connection (in the case of wall connected with a window—separately for the external wall and separately for the window; in the case of connection of the external wall with the ceiling—separately for the lower and upper part of the connection) we can determine additional losses in relation to individual envelopes occurring in the 2D connection.
- The analysis of physical parameters of the connections requires individual computer simulation, taking into account variable structural and material solutions (especially thermal insulation) as well as the parameters of indoor and outdoor air. This leads to the optimal selection of the material solution (before and after insulation) in terms of hygrothermal conditions.

**Author Contributions:** Conceptualization, P.K.; formal analysis, A.N.; funding acquisition, K.P.; investigation, P.K., A.N.; methodology, P.K.; resources, P.K.; software, K.P.; supervision, A.N.; visualization, P.K., K.P.; writing—original draft, A.N.; writing—review and editing, A.N. All authors have read and agreed to the published version of the manuscript.

**Funding:** This research received no external funding.

**Conflicts of Interest:** There is no conflict of interest.

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
