# Peer review of "The Impact of Internal Insulation on Heat Transport through the Wall: Case Study"

_applsci, doi:10.3390/app10217484_

Round 1
Reviewer 1 Report
General comments
The paper describes the influence of the insulation of the inner side of an institutional building, in term of thermal transmittance and humidity level. First, experiments were carried out in the building and numerical simulations were performed in a second time.
Overall, the paper offers a practical analysis of the influence of inner thermal insulation (when an external insulation is not applicable), both on the thermal performances and the humidity level. The experimental part is of interest, well described and provides qualitative and quantitative results.
However, the numerical part lacks of proper description for the case study (what does the geometry look like), do not provide any explanations on the FEM solver, neither on the boundary conditions (I guess fixed temperature), which makes the results somehow confusing.
While the experimental part is worthy of publication I do not think the numerical part reached the standard of publication. Therefore I recommend major revision of the numerical part before acceptance.
Detailed comments
Introduction:
- I do not understand the link between the temperature measurements of building and the impact of connectors with metal pins in the sentence: “Thermal performance of buildings can be determined by temperature measurements [11], and in such cases the impact of connectors with metal pins [12] cannot be ignored”. Is it the change in emissivity due to different materials?
- In the sentence: “Unfortunately, when insulating from the inside, apart from the risk of condensation, there are also other problems [30] related primarily to thermal bridges that are difficult to avoid [31], [32].” It would be interested to list the problems, since those are unknown to anyone who is not expert in thermal insulations.
The introduction is very compact and the referred papers are just listed, without further explanations. Therefore, it is difficult to distinguish between a proper literature review or just name dropping. I do not ask the authors to change it in the current study but they should be aware of it.
Results:
- Figure 6: Why not plotting the lines Li1 and Li3 (and also Li2 and Li4) in the same graph? It would help when comparing the results.
- Figure 7: Consider changing the temperature legend to a common one (such as the one used in Figure 4).
- “Additionally, a decrease in temperature on the surface of the insulation is observed at the place of the existing mechanical connector” Isn’t it due to the change in emissivity?
- Table 4: It would help the reader to remind the theoretical values of the thermal transmittance of table 1 and 2 in table 4, as well as their respective discrepancies.
- Figure 8: Please specify the units of the interval of recordings in the figure.
- There is no info of the relationship used to calculate L2D.
- Figure 9: It is very difficult to understand what is represented, portions of walls close to the window? A proper definition of the studied geometry (maybe a 3D sketch) with corresponding boundary conditions, maybe a reference frame, will help to visualize what part of the walls are shown. Overall the simulation part needs a better structure, the results seem to appear without much explanations.
Author Response
The paper describes the influence of the insulation of the inner side of an institutional building, in term of thermal transmittance and humidity level. First, experiments were carried out in the building and numerical simulations were performed in a second time.
Overall, the paper offers a practical analysis of the influence of inner thermal insulation (when an external insulation is not applicable), both on the thermal performances and the humidity level. The experimental part is of interest, well described and provides qualitative and quantitative results.
However, the numerical part lacks of proper description for the case study (what does the geometry look like), do not provide any explanations on the FEM solver, neither on the boundary conditions (I guess fixed temperature), which makes the results somehow confusing.
While the experimental part is worthy of publication I do not think the numerical part reached the standard of publication. Therefore I recommend major revision of the numerical part before acceptance.
Answer:
Thank you for your valuable comments. We have tried to include all of them in our work. 3.2. Simulations
Detailed comments
Introduction:
- I do not understand the link between the temperature measurements of building and the impact of connectors with metal pins in the sentence: “Thermal performance of buildings can be determined by temperature measurements [11], and in such cases the impact of connectors with metal pins [12] cannot be ignored”. Is it the change in emissivity due to different materials?
Answer:
Thank you for your comment. We have added the following in the text line 33-34
“The said connectors have much higher heat conduction than the structural wall or the insulation, and therefore they have impact on the rise of heat loss of the building [6].”
- In the sentence: “Unfortunately, when insulating from the inside, apart from the risk of condensation, there are also other problems [30] related primarily to thermal bridges that are difficult to avoid [31], [32].” It would be interested to list the problems, since those are unknown to anyone who is not expert in thermal insulations.
Answer:
We have added the following sentence: (line 61-63
Such problems can comprise: mold development, biological corrosion, the decrease of thermal resistance and increased heat loss.
The introduction is very compact and the referred papers are just listed, without further explanations. Therefore, it is difficult to distinguish between a proper literature review or just name dropping. I do not ask the authors to change it in the current study but they should be aware of it.
Answer:
Thank you for your comment. Of course, we will follow the reviewer's advice in our future studies.
Results:
- Figure 6: Why not plotting the lines Li1 and Li3 (and also Li2 and Li4) in the same graph? It would help when comparing the results.
Answer:
We have corrected it in compliance with the reviewer's comment
- Figure 7: Consider changing the temperature legend to a common one (such as the one used in Figure 4).
Answer:
Figure 7 was prepared so that the place where the network of connectors is located could be seen. That is why it has such a form. We have improved the legend.
- “Additionally, a decrease in temperature on the surface of the insulation is observed at the place of the existing mechanical connector” Isn’t it due to the change in emissivity?
Answer:
We have added the following sentence: (line 223-225)
This may be caused by the change of heat conductivity coefficient, although a slight influence of the change in emissivity at this place cannot be ruled out.
- Table 4: It would help the reader to remind the theoretical values of the thermal transmittance of table 1 and 2 in table 4, as well as their respective discrepancies.
Answer:
We have complemented Table 4
- Figure 8: Please specify the units of the interval of recordings in the figure.
Answer:
We have complemented it
- There is no info of the relationship used to calculate L2D.
Answer:
It has been added in chapter 3.2. Simulations
- Figure 9: It is very difficult to understand what is represented, portions of walls close to the window? A proper definition of the studied geometry (maybe a 3D sketch) with corresponding boundary conditions, maybe a reference frame, will help to visualize what part of the walls are shown. Overall the simulation part needs a better structure, the results seem to appear without much explanations.
Answer:
It has been complemented in chapter 3.2. Simulations
Reviewer 2 Report
This paper is a detailed technical report about internal insulation in a building which did not admit outer in sublation for aesthetic reasons. Though English must be improved, in particular in the introduction section, it is well written and comprehensible. It is hard for me to evaluate the scientific relevance of this case study. I believe that however it is of interest for civil engineers and practitioners in the field of thermal insulation.
Author Response
Thank you for reading our article. The introduction has been supplemented as indicated by the reviewers. The correctness of English was verified by a language expert.
Reviewer 3 Report
This manuscript Paweł Krause et al. presents a case study on how to improve the energy efficiency of a building of significant heritage value through retrofitting the external wall system in Upper Silesia, Poland, based on measurements (quantitatively by means of heat flux measurement and qualitatively by means of infrared temperature measurement) and simulations. The methods used for simulation are reasonable, and well executed. Important standard thermal parameters have been extracted for thermal performance comparisons. Later, they added more simulation on thermal bridges to investigate energy improvements.
1-Introduction should mention the study of thermal bridges
2- Interior conditions should be defined more the: is there a heating system in the interiors? What are its operating parameters?
3- Lines 167-170, Indicate where this reference percentage was obtained from: standard, bibliographic reference, etc
4-Figura 4. There are two thermal imaging images. It would be interesting to see the actual reference image
5- In formula (2), line 254-255, describe each acronym
Author Response
Thank you for your valuable comments that will help to improve our article. The additions are marked in red in the article.
This manuscript Paweł Krause et al. presents a case study on how to improve the energy efficiency of a building of significant heritage value through retrofitting the external wall system in Upper Silesia, Poland, based on measurements (quantitatively by means of heat flux measurement and qualitatively by means of infrared temperature measurement) and simulations. The methods used for simulation are reasonable, and well executed. Important standard thermal parameters have been extracted for thermal performance comparisons. Later, they added more simulation on thermal bridges to investigate energy improvements.
1-Introduction should mention the study of thermal Bridges
Answer:
Thank you for your comment. We have included it in the Introduction.
2- Interior conditions should be defined more the: is there a heating system in the interiors? What are its operating parameters?
Answer:
It is in the text line 245-250
3- Lines 167-170, Indicate where this reference percentage was obtained from: standard, bibliographic reference, etc
Answer:
We have complemented it. It is in ISO 9869 [43]
4-Figura 4. There are two thermal imaging images. It would be interesting to see the actual reference image
Answer:
We have complemented the sentence, specifying that the actual image is in Fig.2. In the text it reads as follows:
“The results of thermal imaging measurements before and after the application of the insulation layer on the inner wall surface (Fig.2) are presented in Fig. 4”
5- In formula (2), line 254-255, describe each acronym
Answer:
We have complemented it
Round 2
Reviewer 1 Report
I want to thank the authors for providing meaningful revisions to their paper according to my remarks. I hope that my comments contributed in a positive way to the quality of this study.
I noticed a significant improvement of the numerical part, with a better understanding of the FEM model and a better description of the case studies. Even so I would have liked a 3D representation of the geometry under investigation, I believe the results are clear enough for readers.
Therefore I recommend this paper for publications.
Two little remarks:
- Equation (3) and (4) are the same. Isn't L2D simply the term on the left of the temperature difference in eq. (2) ?
- The lines 363 to 368 is a repetition of the paragraph line 317 - 322.
Author Response
Thank you very much for your comments. Thank you for accepting the article. We have 2D analysis software. We will have new software for 3D analysis soon. Then our work will be better.
- Equation (3) and (4) are the same. Isn't L2D simply the term on the left of the temperature difference in eq. (2)?
Answer:
We improved
- The lines 363 to 368 is a repetition of the paragraph line 317 - 322.
Answer.
we removed the redundant paragraph line 317-322